# Fouling Control Strategies for High Concentrated Liquid Desiccants Concentrating Using Membrane Distillation

**DOI:** 10.3390/membranes13020222

**Published:** 2023-02-10

**Authors:** Seonguk Ha, Jieun Lee, Seongeom Jeong, Sanghyun Jeong

**Affiliations:** 1Department of Environmental Engineering, Pusan National University, Busan 46241, Republic of Korea; 2Institute for Environmental and Energy, Pusan National University, Busan 46241, Republic of Korea

**Keywords:** fouling control, lithium chloride, liquid desiccant, membrane distillation, potassium formate

## Abstract

Air conditioning using a liquid desiccant (LD) is an energy-efficient air purification and cooling system. However, high energy is required to concentrate or regenerate the LD. This study aimed to investigate the characteristics of membrane fouling in more detail and determine control strategies for LD concentrating using membrane distillation (MD). Two different LDs—lithium chloride (LiCl) and potassium formate (HCOOK)—were used. Because LDs require high concentrations by nature (i.e., 40 wt% for LiCl and 70 wt% for HCOOK), the concentration was started from half of those concentrations. This resulted in a flux decline with severe membrane fouling during the concentration using MD. Different membrane fouling mechanisms were also observed, depending on the LD type. Three different physical membrane fouling control methods, including water flushing (WF), air backwashing (AB), and membrane spacer (SP), were introduced. Results showed that WF was the most effective. Both AB and SP showed a marginal change to no cleaning; however, an initial flux with SP was about 1.5 times higher than no cleaning. Therefore, WF combined with the SP could maintain a high flux and a low fouling propensity in the treatment of a high-concentration solution using MD.

## 1. Introduction

The liquid desiccant air conditioning (LDAC) system has recently been acknowledged as an energy-saving technology that controls air conditioning space loading for cooling [1]. Conventional air conditioning or dehumidification systems consume unnecessarily high electrical energy because of their operating principle. They overcool the air below the dew point to remove moisture and then heat it again to an appropriate temperature indoors. However, LDAC does not need to be overcooled because the liquid desiccant (LD) has hygroscopicity by itself, which is the property of absorbing moisture [2,3]. The LD absorbs moisture from the air and requires a regeneration step to continuously operate the LDAC process. To design an efficient LDAC process, the regeneration of a used (or diluted) LD is important. Because regeneration accounts for up to 75% of the total energy consumption in the LDAC system [4], optimizing its efficiency is the key to reducing the overall operating costs.

Various types of LD have inherent properties. To effectively utilize them in LDAC systems, the following properties should be guaranteed: low vapor pressure, high solubility, low viscosity, low corrosiveness, high chemical stability, nontoxicity, non-volatility, and the absence of odor [5]. Lithium chloride (LiCl) has been reported as one of the superior LDs, owing to its relatively low vapor pressure [5]. However, its use in LDAC systems has some drawbacks; it can damage equipment because of its high corrosiveness and its price is relatively high. In contrast, potassium formate (HCOOK) is an alternative to LiCl with a similar vapor pressure. In addition, its price and corrosiveness are significantly lower than those of LiCl. Therefore, HCOOK is expected to be more efficient than LiCl for LDAC [6].

Membrane distillation (MD) is a thermally driven membrane process that uses a microporous hydrophobic membrane and temperature differentials to achieve product concentration through evaporation and condensation [7]. It can extract water for reuse and concentrate products. Therefore, MD can be used for LD regeneration by concentrating only the water from the diluted LD. Moreover, as only vapor can penetrate the hydrophobic membrane, MD can theoretically reject salts containing LD, such as LiCl and HCOOK [8], making LD loss negligible during LD regeneration by MD. As the regeneration of LDs is the main process of the LDAC, various methods have recently been studied. Su et al. [9] reported that various types of LD regeneration were conducted for energy saving and optimizing processes (i.e., solar thermal driven regeneration systems, heat pump driven regeneration systems, photovoltaic/thermal hybrid, photovoltaic-electrodialysis, ultrasonic atomization-thermal hybrid LD regeneration systems). The MD process can also be operated at a moderate feed temperature with low-grade waste heat and solar thermal energy [10], which can be helpful in reducing the energy cost for LD regeneration in LDAC systems.

The major problem in membrane processes, such as MD, is the rapid decline of permeate flux over time due to membrane fouling [11]. A decrease in flux indicates a reduction in the efficiency of MD. When fouling occurs on the membrane, the temperature polarization (TP) between the feed and permeate flux reduces the driving force across the membrane [12,13]. Consequently, the energy consumption and the time of LD concentration increase, and membrane fouling and wetting can be significantly accelerated. Therefore, membrane fouling must be controlled because a stable flux is an important factor in the MD process [14].

Fouling is classified as inorganic, organic, biological, and colloidal. LD causes inorganic fouling, which is the scaling formation of inorganic salts on the surface or inside a membrane [15]. Foulants accumulate in a membrane by concentrating LiCl, and the HCOOK can be deposited on the membrane surface or block the pores that cause thermal and hydraulic resistance [16]. Inevitably, LD must be driven at a high concentration to have a high water sorption capacity and then it should be highly concentrated for the LDAC system; so, MD must be highly re-concentrated. Therefore, membrane cleaning or fouling control steps should be conducted to achieve effective LD concentration or recovery.

There are three main types of physical cleaning methods for mitigating membrane fouling: hydraulic, pneumatic, and sonication. Hydraulic cleaning includes forward and backward flushing, which swipes away foulants from the membrane surface with a liquid, such as deionized (DI) or produced water. Pneumatic cleaning is cleaning with air, so a dried membrane can prevent membrane wetting. Julian et al. [17] reported that vacuum MD and crystallization with periodic air backwash could remove crystal deposition to mitigate membrane wetting. Sonication (or ultrasound) provides cavitation that breaks the concentration polarization (CP) and cake layer, but Guo et al. [18] reported that it has a critical drawback in that crystal formation by nucleation hinders the cleaning of fouling.

Water flushing is the process of washing a membrane with flowing water at a regular frequency and duration. By frequently flushing the membrane with water, the induction time for crystal formation can be reset and severe scaling can be prevented. Zou et al. [19] reported that CaSO_4_ crystals accumulated on the membrane surface could be removed with only a small amount of residual crystals by simple water flushing, without additional chemicals. After the offline water treatment, the permeate flow rate was restored to the initial flux. Air backwashing reduces the contaminated layer by utilizing the shear force exerted on the feed side of the membrane by the air bubbles. Its effectiveness parameters are frequency, duration, and air pressure. Ye et al. [20] reported that disturbing the redeposition of fouling materials in an effective filtration zone is also an effective factor. Recently, the use of feed spacers has attracted attention as an effective method for improving flux and removing fouling by increasing the turbulence of the feed flow. Alwatban et al. [21] reported that both CP and TP were mitigated significantly, and the water flux was enhanced by approximately 40% with a module containing embedded spacers of the larger strand.

Therefore, the objective of this study is to investigate membrane fouling during the regeneration of high concentrations of LD in detail based on the flux pattern of each concentrated LD to establish fouling control strategies. Three types of fouling control methods (water flushing, air backwashing, and feed spacer) were tested, and the fouling reduction effect was analyzed in terms of a stable LD concentration with fouling control in the MD operation. To the best of our knowledge, no study has focused on such a high concentration (i.e., up to 70 wt% for HCOOK) using MD, so this study will provide a key strategy to control fouling at high concentrations using MD. Additionally, it can help to effectively regenerate LD to save energy in LDAC systems.

## 2. Materials and Methods

### 2.1. Materials

#### 2.1.1. Liquid Desiccants

Two LDs, LiCl (lithium chloride, 99%) and HCOOK (potassium formate, 99%), were procured from Sigma-Aldrich (St. Louis, MO, USA). The initial concentration of both LDs was set to half the optimal concentration to be used in LDAC; therefore, for LiCl, the initial concentration was 20 wt% and that for HCOOK was started at 35 wt% [6,12], and they were concentrated to 40 wt% and 70 wt% for LiCl and HCOOK, respectively. In other words, as LDAC systems require a very high concentration of LD for dehumidification, two different LDs were concentrated until the concentration factor (CF) reached 2 for LDAC.

#### 2.1.2. Membrane

A 0.22-μm pore-sized hydrophobic polyvinylidene fluoride (PVDF) membrane (GVHP14250, Durapore, Germany) was used. The thickness and porosity of the membrane were 125 μm and 75%, respectively.

#### 2.1.3. Direct Contact Membrane Distillation (DCMD) Set-Up

The LDs were concentrated in a laboratory-scale DCMD setup device (Figure 1). The channel length, width, and depth of the DCMD module were 65 mm, 15 mm, and 20 mm, respectively. The active membrane area was 975 mm^2^. The acrylic block length, width, and depth of the DCMD module were 130 mm, 85 mm, and 20 mm, respectively. The feed stream consisting of LDs was heated to three different temperatures (60, 70, and 80 °C) with a hotplate stirrer. It was then stirred at 240 rpm to prevent the LD crystals from settling in the feed bath. The permeate stream was cooled constantly at 20 °C by a chiller to condense the distilled water vapor in the permeate bath. The flow rates of the feed and permeate streams were set to 1.0 L min^−1^. It was predetermined to have sufficient turbulence and heat exchange rate [22]. Prior to the LD concentration treatment test, the DCMD baseline test was performed with deionized (DI) water for 30 min at set temperatures. The baseline water flux was used as the initial flux assessment for all further experiments with the LD. All tests were conducted at least three times for reproducibility.

### 2.2. Membrane Fouling Test

#### 2.2.1. Membrane Fouling Development

As stated in Section 2.1.1, MD fouling tests were conducted from half the optimal concentration of each LD to reach a weight concentration factor (CF) of 2. In this case, the vapor pressures of the two LD types were similar during the MD operation. During concentration by MD, the permeate flux decline pattern was observed to determine the fouling mechanism. In addition, the feed and permeate concentrations were monitored to evaluate LD regeneration efficiency using conductivity measurements.

#### 2.2.2. Fouling Mechanism Analysis

Membrane filtration was performed in a constant transmembrane pressure mode with a feed flow normal to the membrane surface and spherical foulants that were completely retained. The equations describing the relationship between the initial flux (*J*_0_), filtration time (*T*), permeate volume of water (*V*), and area of membrane (*A*) for each of the four fouling mechanisms are as follows (Equations (1)–(4)). *K_b_*, *K_s_*, *K_i_*, and *K_c_* represent the complete, standard, intermediate blocking, and cake filtration kinetic rates for the fouling mechanisms, respectively. Each fouling event is shown in Figure 2 [23].
(1)$$\frac{d\left(V/A\right)}{dT}=J_{0}-K_{b}\left(V/A\right)\;\;(\mathrm{Complete}\mathrm{blocking})$$
(2)$$\frac{T}{V/A}=\frac{1}{J_{0}}-K_{s}\frac{T}{2}\;\;(\mathrm{Standard}\mathrm{blocking})$$
(3)$$\frac{dT}{d\left(V/A\right)}=\frac{1}{J_{0}}-K_{i}T\;\;(\mathrm{Intermediate}\mathrm{blocking})$$
(4)$$\frac{T}{V/A}=\frac{1}{J_{0}}-K_{c}\frac{V/A}{2}\;\;(\mathrm{Cake}\mathrm{filtration})$$

In the MD operation concentrating on the LDs, we determined the fouling mechanisms based on fouling patterns.

### 2.3. Membrane Fouling Control Strategies

#### 2.3.1. Water Flushing

The purpose of water flushing (WF) is to remove a constructed layer of contaminants on the membrane through the creation of turbulence (Figure 3b). A high hydraulic pressure gradient was used during WF, which was conducted for 1 min at 1 h intervals with a 2.5 L min^−1^ flowrate.

#### 2.3.2. Air Backwashing

Air backwashing (AB) was used to remove fouling from the membrane pores and was conducted for 1 min every 1 h with an air pressure of 100 kPa. Air was pumped on the permeate side in the DCMD module to push out LD salts that were stuck in the membrane pores. As one of the proper functions of AB is to dry out a membrane, it can resist membrane wetting (Figure 3c).

#### 2.3.3. Membrane Feed Spacer

The two main functions of membrane spacers (SP) are to fix the position of the membrane, which could prevent membrane deformation and enable efficient mass exchange between the bulk stream and membrane surface. The improvement in membrane performance was induced by reducing fouling and improving the flux owing to the increasing effect of turbulence. As shown in Figure 3d, an SP made of 45° woven square patterned polylactic acid (PLA) was designed to cover all the effective membrane surfaces.

### 2.4. Analytical Methods

#### 2.4.1. MD Flux and Its Decline


(5)
$$J=\frac{\Delta W}{\Delta t\times A}$$


To analyze the efficiency of LD regeneration by MD, the water flux was calculated according to Equation (5). The water flux was expressed as *J* (L m^−2^ h^−1^, LMH), where *L* is the weight of the permeate water vapor, *T* is the unit of time in h, and *A* is the effective membrane area. The higher the change in weight over the area and time is, the greater the penetration of water through the membrane. Therefore, the flux can be considered a parameter of the MD efficiency.
(6)$$\mathrm{FDn}\left(\%\right)=\left(1-\frac{J_{f}}{J_{0}}\right)\times 100$$

The normalized flux decline (FDn) was calculated to compare the decrease in flux between the fouling control methods (Equation (6)), where *J*_0_ and *J_f_* are the initial and final fluxes, respectively. This value can also indicate a decline in membrane performance due to membrane fouling.

#### 2.4.2. LD Rejection Efficiency


(7)
$$\mathrm{Rejection}\;\mathrm{efficiency}\;\left(R,\;\%\right)=\left(\frac{C_{p}\times V_{p}}{C_{f}\times V_{f}}\right)\times 100\%$$


Inductively coupled plasma optical emission spectrometry (ICP-OES, Optima 8300, PerkinElmer, Waltham, MA, USA) was used to measure the LD concentrations. The LD rejection efficiency was calculated using Equation (7) and indirectly provides information on how well LDs are concentrated during the MD operation. *C_p_* and *C_f_* are the concentrations (mg/L) of the permeate and feed solutions, respectively, and *V_p_* and *V_f_* are the volumes (L) of the permeate and feed solutions, respectively. We compared these values at CF 1.0 and 2.0 for both LiCl and HCOOK. Since the MD process passes only water vapor through the membrane by the hydrophobic membrane, it is theoretically 100% rejection of salt-like LD. However, there is a possibility that partial or full wetting or fouling have occurred.

#### 2.4.3. Membrane Fouling Characterization

Scanning electron microscopy with energy dispersive spectroscopy (SEM-EDS) (ZEISS, Jena, Germany) was used to study the surface and cross-sectional morphologies of both clean and fouled membranes by elemental analysis. This analysis was used to identify fouling behavior based on the flux pattern. The water contact angle was measured using a goniometer (SEO, Gyeonggi, Republic of Korea) and used to determine the hydrophobicity change before and after the concentration of LDs. Water droplets were dropped on the feed side of the membrane surface, and each sample was analyzed at least three times. The average values are reported in this paper.

## 3. Results and Discussion

### 3.1. LD Concentration by DCMD

#### 3.1.1. Effect of Feed Temperature in DI Water

To evaluate the DCMD performance, the system was operated under different feed temperatures, as the main driving force of the MD is temperature. Figure 4 shows the baseline test results with DI water and the flux pattern (behavior) as a function of temperature for 24 h.

The DI water fluxes at 60, 70, and 80 °C of feed temperature with a constant permeate temperature (20 °C) were 19.67 (± 1.33), 27.03 (± 0.95), and 46.98 (± 2.11) LMH, respectively. The flux at a feed temperature of 80 °C was more than two times higher than that at 60 °C, indicating that temperature is the main factor in the MD process. In fact, the temperature is a key parameter for designing engineering processes by comparing performance and cost. The flow rate was adjusted to 1.0 L min^−1^ because the DI water flux in the MD system was stable under these conditions.

#### 3.1.2. The Effect of Feed Temperature in LDs

LD regeneration tests were conducted, and fluxes according to time and membrane fouling were analyzed. Before conducting a legitimate test, the membrane was subjected to DI water distillation for 30 min to stabilize the membrane in the module and facilitate reproducible research. Again, as the optimum LD concentrations are approximately 70 wt% and 40 wt% for HCOOK and LiCl, respectively, LDs were regenerated from half of the optimal concentration, assuming a situation that absorbs moisture in the air [24]. Therefore, HCOOK (35 wt%) and LiCl (20 wt%) were concentrated with DCMD at feed temperatures of 60, 70, and 80 °C with a 1 L initial feed solution volume (Figure 5).

As shown in Figure 5a,b, severe flux decline occurred at the highest temperature (80 °C) tested in both HCOOK and LiCl after 2 h. At 60 °C, the regeneration time was 71.03 h for HCOOK and 93.17 h for LiCl, which was too long for regeneration. FDn was 84.61% for HCOOK and 81.72% for LiCl, so the flux was relatively constant compared to that with the other temperature conditions (70 and 80 °C). However, the flux was very low, making it unsuitable for LD regeneration. However, the 80 °C condition showed a different result. The initial fluxes were the highest, 30.90 LMH for HCOOK and 27.49 LMH for LiCl, and the concentration time was relatively short. However, the flux reduction rate was very high, as shown in Figure 5; therefore, the regeneration process was unstable. Furthermore, there is concern that the possibility of membrane damage may increase. However, at 70 °C, the initial fluxes of LDs were 21.34 LHM for HCOOK and 20.34 LMH for LiCl, and FDn values were 71.62% for HCOOK and 73.89% for LiCl. Based on the results, we concluded that the difference between HCOOK and LiCl was not significant, and the concentration time was not significantly different from that at 80 °C. Therefore, in this study, it was determined that 70 °C was the most suitable temperature condition for the LDMD system. Additionally, it may be efficient in terms of energy savings.

### 3.2. MD Performance Analysis with Flux and LD Concentration

Flux is a basic performance parameter for a process using a membrane, and in this study, the higher the flux was, the shorter the concentration time. This implies that the concentration was completed with less energy, which directly affects the efficiency of the process. Analyzing the LD concentration shows the LD loss amount and how the purity of the distilled water produced and measuring conductivity can provide information on whether the LD is well-concentrated and whether it is not transferred to the distilled permeate tank.

Figure 6 shows the flux decline according to CF, indicating how LDs were concentrated. Without any other controlling methods (no cleaning), the flux when HCOOK was concentrated from 35 to 70 wt% was relatively high, meaning that HCOOK was concentrated better than LiCl during the same time and the concentration was even higher compared to that of LiCl. With WF, an improved flux could be obtained than with no cleaning. It was possible to operate with low flux reduction, which means the foulant was eliminated effectively and LD could be concentrated at a high and steady flux. However, AB had little effect on cleaning the foulant. The cause of the flux reduction in AB was due to the reduction in membrane temperature during the AB. In practice, maintaining temperature during the MD cleaning is one of the significant points. In terms of flux recovery, it was not effective in preventing the membrane from wetting or removing the foulant inside the membrane.

The membrane SP showed remarkable flux improvement. The initial flux appeared to have nearly doubled compared to that without cleaning. However, at the endpoint of the concentration, the results were similar to those of no cleaning, revealing that the fouling formation increased with a high loading rate of foulant due to the high flux. Nevertheless, the high flux achievement showed that the turbulence caused by the membrane SP alleviated the formation of fouling during the LD concentration.

All the flux results showed that WF was the most effective in the high concentration of the LD solution using MD. This indicates that WF can control the fouling and increase the efficiency of the MD operation.

### 3.3. Fouling Behavior and Its Mechanism

#### 3.3.1. Membrane Fouling Mechanism

The fouling mechanisms of different LDs on the membrane in the MD concentration were estimated by matching the flux pattern with that calculated using Equations (1)–(4). As shown in Figure 7, LiCl and HCOOK were fouled differently on the MD membrane. HCOOK was dominantly fouled by cake filtration, but it was weakened to intermediate and standard blocking with cleaning. For LiCl, the fouling mechanism changed from standard to intermediate blocking with no cleaning, and from the inside to the surface. However, the fouling effect deteriorated by maintaining standard blocking with cleaning. Specific discussions are presented in the following sections.

#### 3.3.2. No Cleaning

LDs were regenerated using MD with no cleaning to accurately observe the effect of cleaning. As shown in Figure 7, the flux decline rate was similar, but the average flux of HCOOK was higher than that of LiCl. This means that the regeneration time for LiCl was approximately 17 h longer. As the fouling of HCOOK was cake formation, but LiCl fouled on the membrane as intermediate and standard blocking, fouling control methods were expected to be differently effective.

#### 3.3.3. Water Flushing

The WF method is generally utilized as a membrane-cleaning tool because of its simple and powerful characteristics. Naidu et al. [25] used WF to efficiently remove foulants for drinking water production. The Mg, Na, and Cl ions could be washed out by periodic DI-WF with vacuum-enhanced multi-effect membrane distillation (V-MEMD). The membrane was washed at a 1 min h^−1^ interval at 2.5 L min^−1^ using 2 L of DI water. This could disturb the crystallization of the membrane by the LDs and induce a constant flux state. The fouling mechanism results indicated that HCOOK was fouled in intermediate blocking and was washed away from the membrane surface as it changed from cake filtration to intermediate blocking. LiCl also changed from intermediate to standard blocking. It appeared that fouling moved from the surface to the interior, but fouling was relatively controlled by the foulant formed on the membrane.

#### 3.3.4. Air Back Washing

AB is a physical cleaning method that is widely used in pressure-driven membrane processes, including ultrafiltration water treatment and membrane bioreactors [14]. Zou et al. [19] suggested that when flux reduction occurred regularly or crystallization was homogeneous, it was effective to remove the foulant. In this case, it was possible to achieve a flux reduction of 54% after 15 h with AB compared to the membrane without any other treatment. Instead, the heterogeneous salt was concentrated, and the flux remained constant only for a short time and dramatically declined. This means that the reduction to flux decline was not effective because it could not eliminate the fouling layer [17]. From the results of the present study, LD fouling was heterogeneous salt crystallization because AB was ineffective in recovering the flux.

#### 3.3.5. Membrane Spacer

An effective advantage of membrane SP is the enhanced performance of the membrane by increasing the flux. This is a remarkable characteristic because it does not require additional energy. In this study, a 45° orientation membrane SP made of PLA was utilized with a thickness of spacer of 1 mm, and a filament length of 5 mm. With the membrane SP, all the LD fouling mechanisms were changed. HCOOK changed from cake filtration to standard blocking and LiCl changed from intermediate to standard blocking. Membrane SP, which has two sets of SP filaments being oriented at an angle of 45° (45° spacer orientation), showed the best experimental results, indicating that the drag coefficient and the Nusselt number were higher than the other comparative experimental group (90°, 62°, 30°, and 45° top filaments orientation). This is a similar result to the present study. Moreover, the results of the three-dimensional computational fluid dynamics (3D CFD) simulation showed the drag coefficients on the Nusselt number and Reynolds number are important for the design of membrane SP [26,27,28,29].

### 3.4. Fouling Analysis

#### 3.4.1. Contact Angle Analysis

To analyze the damage to the hydrophobic membrane, the water contact angle data is commonly used because it measures the loss of hydrophobicity of the membrane. In all cases, HCOOK caused less membrane damage than LiCl (Figure 8). This means that HCOOK has a superior capability for the membrane because it can operate for a long time with a lower possibility of membrane damage.

In physical cleaning, WF was the most effective method for maintaining hydrophobicity. Thus, it seems that the WF method was able to reduce the damage to the membrane by directly cleaning the surface.

#### 3.4.2. LD Concentration

By comparing the start and end points of regeneration, we determined how LDs were concentrated without loss. All rejection efficiencies were higher than 99.98%, indicating that the membrane functioned sufficiently as a semi-permeable barrier.

#### 3.4.3. Optical Results with SEM and SEM-EDS Analysis

SEM analysis revealed that the LDs affected the membrane pore structure. As shown in Figure 9, the foulant layer covered the membrane, and the LD interrupted the passage of steam in the pores under all temperature conditions, as expected from the fouling mechanism estimation. Foulants were heavily dominated by the membrane pore structure, thus confirming that a high concentration of LDs led to severe membrane fouling that required membrane cleaning.

The SEM-EDS results were utilized to analyze the accumulation of LDs on the surface and inside the membrane. As shown in Figure 10, HCOOK was fouled as cake filtration, and LiCl was fouled as intermediate blocking. In the analysis, the distribution of LDs was visually recognized by expressing different colors for each chemical element (C, O, K, and Cl), while F (one of the PVDF membrane elements) is expressed in green. Both LDs were evenly distributed on the membrane surface, and the analysis of the cross-section in the membrane confirmed that HCOOK easily penetrated and was distributed evenly inside the membrane, but LiCl rarely penetrated and was distributed thickly only on the membrane surface. Thus, whether cleaning methods could effectively remove fouling could be determined by comparing the difference between the SEM-EDS results before and after cleaning (Figure 9 and Figure 10).

As shown in Figure 10, less fouling was formed with WF, as expected. However, in the AB result, the LDs still penetrated the inside of the membrane and had no significant effect. It was confirmed that a large amount of foulant accumulated on the surface and inside, owing to the high flux with SP. This shows that optical analysis enabled us to see the effectiveness of physical cleaning methods. In short, WF washed out the foulant effectively.

## 4. Conclusions

The most efficient method for LDAC optimization is reducing the energy of the regeneration process. To achieve low-energy and highly efficient regeneration or concentration, the MD process, which is capable of distillation at mild temperatures, was applied. In particular, various physical fouling control methods were tested, and the main findings are as follows:Membrane SP showed the highest flux result; however, the flux reduction rate was high compared to no cleaning and AB. However, WF could maintain the flux stably.The WF method was the most effective for inhibiting membrane damage and reducing membrane fouling based on rejection efficiency, fouling mechanism, contact angle, and SEM-EDS.In this study, if the WF method is operated with the SP, a higher flux could be maintained in a longer time and concentrated LDs in a shorter time.Regarding energy consumption, WF involves higher pumping energy but for a short time (1 min per h). However, marginal energy is required in other cleaning methods (AB and SP). It is clear that energy saving is mainly attributed to long-term operations with efficient cleaning.

## Figures and Tables

**Figure 1 membranes-13-00222-f001:**
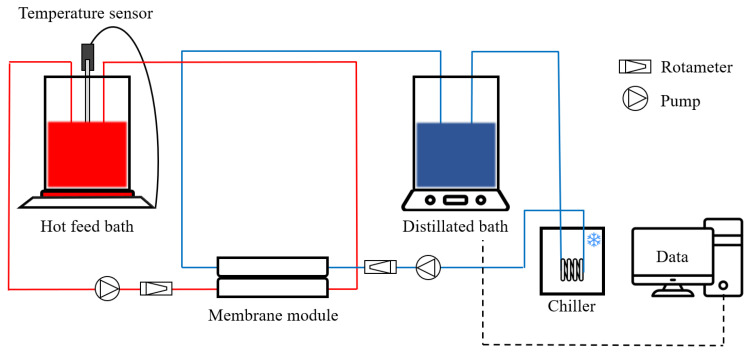
Schematic diagram of DCMD regenerating the LD.

**Figure 2 membranes-13-00222-f002:**
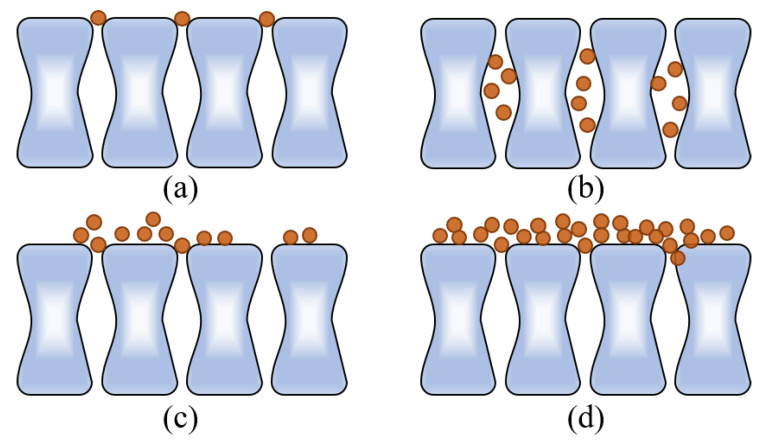
Four types of membrane fouling models. (**a**) Complete blocking, (**b**) standard blocking, (**c**) intermediate blocking, and (**d**) cake filtration.

**Figure 3 membranes-13-00222-f003:**
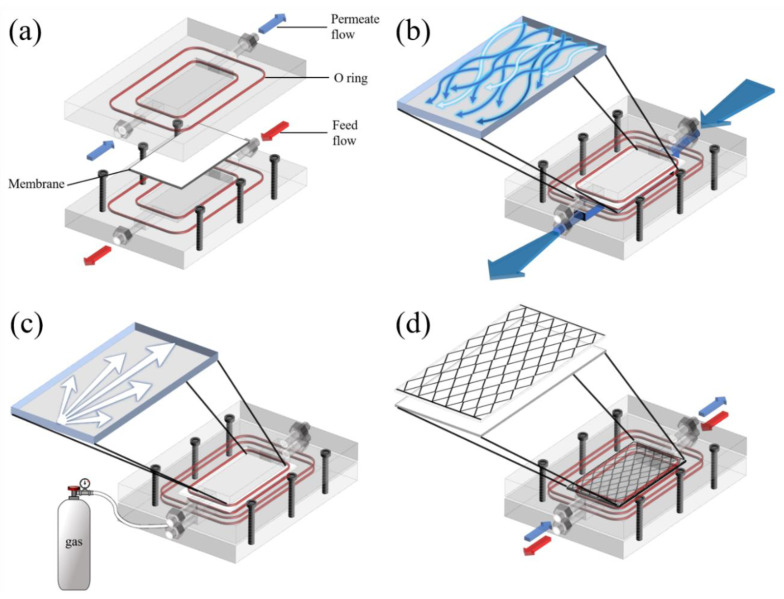
Schematic diagrams of (**a**) the DCMD module and fouling control methods; (**b**) water flushing was conducted for 1 min at 1 h intervals with a 2.5 L min^−1^ flowrate; (**c**) air backwashing was performed for 1 min every 1 h with 100 kPa of air pressure; and (**d**) a membrane spacer was made of 45° woven square patterned PLA.

**Figure 4 membranes-13-00222-f004:**
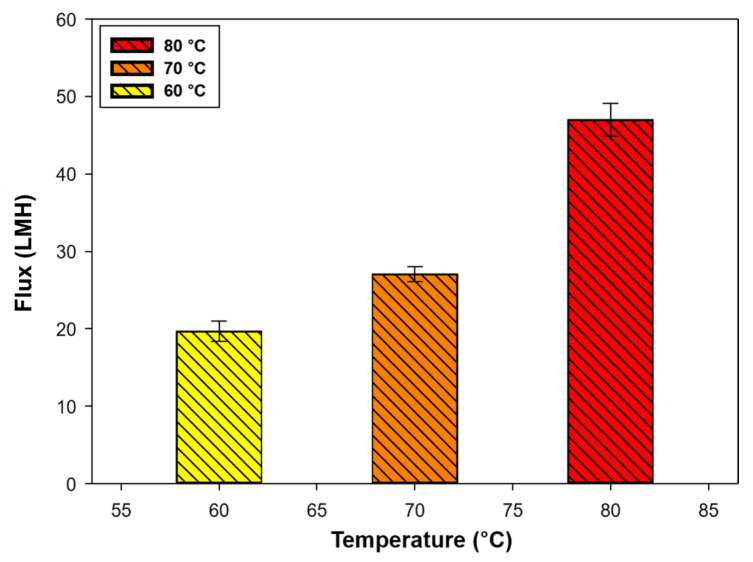
Baseline test results. Water fluxes at feed temperatures of 60, 70, and 80 °C (permeate temperature was maintained at 20 °C, and feed and permeate flow rates were 1.0 L min^−1^).

**Figure 5 membranes-13-00222-f005:**
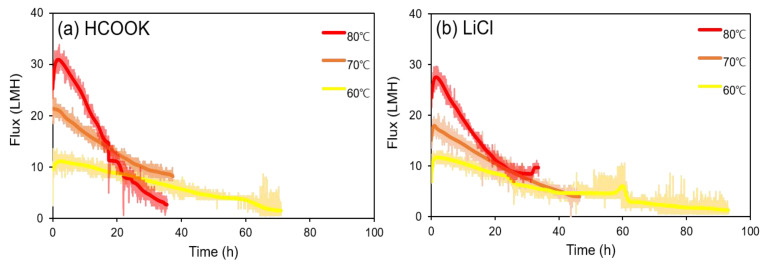
Comparison of flux decline (in terms of concentrating time) depending on the feed temperature of (**a**) HCOOK and (**b**) LiCl.

**Figure 6 membranes-13-00222-f006:**
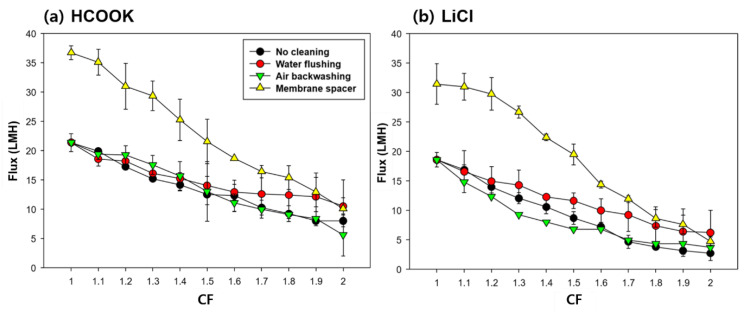
Flux decline patterns as a function of CF when DCMD was operated with different fouling strategies in the concentrating of (a) HCOOK and (b) LiCl (feed volumes of HCOOK and LiCl were 0.65 L and 0.80 L, respectively. The feed temperature was 70 °C. HCOOK and LiCl were concentrated from 35 wt% to 70 wt% and 20 wt% to 40 wt%, respectively).

**Figure 7 membranes-13-00222-f007:**
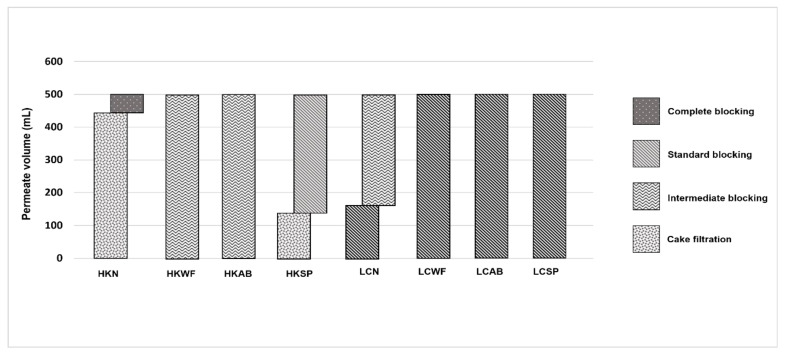
The fouling mechanism data in terms of permeate volume with no cleaning and physical cleaning methods. LDs were expressed as HCOOK to HK and LiCl to LC. LDs with cleaning methods were abbreviated to HKN for HK with no cleaning; HKWF for HK with WF, HKAB for HK with AB, and HKSP for HK with SP. Similarly, LCN for LC with no cleaning, LCWF for LC with WF, LCAB for LC with AB, and LCSP for LC with SP.

**Figure 8 membranes-13-00222-f008:**
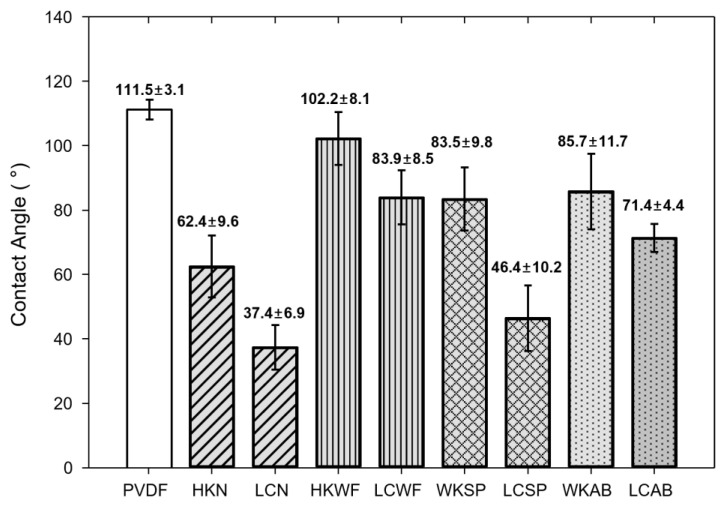
Contact angle results. PVDF is the CA result of the virgin membrane before MD operation. LDs were expressed as HCOOK to HK and LiCl to LC. LDs with cleaning methods were abbreviated to HKN for HK with no cleaning, HKWF for HK with WF, HKAB for HK with AB, and HKSP for HK with SP. Similarly, LCN for LC with no cleaning, LCWF for LC with WF, LCAB for LC with AB, and LCSP for LC with SP.

**Figure 9 membranes-13-00222-f009:**
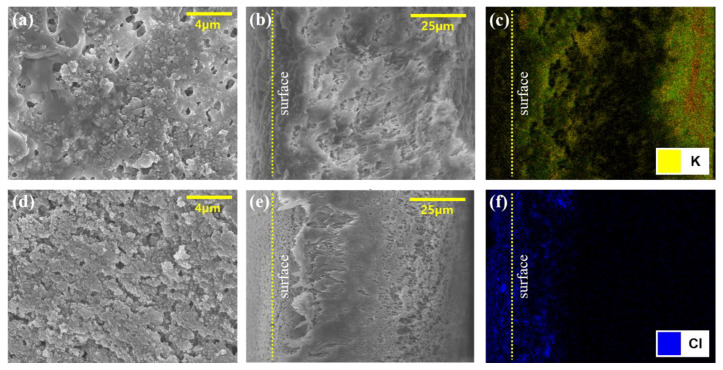
An SEM and SEM-EDS image of LiCl and HCOOK after no cleaning concentration. (**a**,**d**) are surface SEM images of HCOOK and LiCl. (**b**,**e**) are cross-sectional SEM images of HCOOK and LiCl. (**c**,**f**) are cross-sectional SEM-EDS images of HCOOK and LiCl.

**Figure 10 membranes-13-00222-f010:**
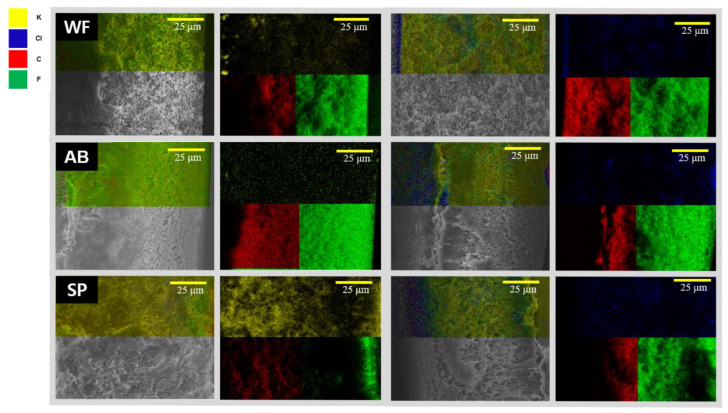
The SEM-EDS results of HCOOK and LiCl on a cross-sectional membrane with the cleaning method. WF: water flushing, AB: air back washing, and SP: membrane spacer.

## Data Availability

The data presented in this study are available on request from the corresponding author.

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
