# Peer review of "Fouling Control Strategies for High Concentrated Liquid Desiccants Concentrating Using Membrane Distillation"

_membranes, 2023, doi:10.3390/membranes13020222_

Round 1
Reviewer 1 Report
The paper entitled ''Fouling control strategies for high concentrated liquid desiccants concentrating using membrane distillation'' aims to investigate different fouling methods for improving MD process performance in concentrating liquid desiccants. Two liquid desiccants, LiCl and HCOOK were tested. Water flushing, air backwashing and membrane feed spacer were implemented to control MD fouling. The authors provided sufficient results to discuss the topic. I believe the manuscript can be accepted for publication after a revision. My comments to further improve the quality of the paper are as follows.
1. The abstract was poorly written. So, it needs to be revised. Please note that abstract should briefly contain the objective, approach (material and methods), key results and implications of the study. Please avoid using general statements in the abstract.
2. Abstract, Lines 12-14: ''Because 12 LDs require high concentrations by nature (i.e., 70 wt% for HCOOK), the concentration was started from half of the optimal concentration.'' This sentence is not clear to me. Please give the concentrations. Please report the key achievements in the abstract.
3. Introduction, Lines 101-102: Please mention which concentrations were tested.
4. Materials and Methods, Section 2.1.1: Please mention the initial and optimal concentrations. Please explain the optimal concentration.
5. Materials and Methods: Please give feed and coolant water volume and MD module dimensions.
6. Results and discussion, Figure 5: The MD experiment durations are too short. The authors should conduct long-term experiments to evaluate the fouling phenomena.
7. Results and discussion, Figure 4: The flux values for DI water can be given in the text. It is not necessary to give them in a figure. Please remove Figure 4.
8. Results and discussion, Section 3.2.1: Plesae mention the initial feed water volume.
9. Results and discussion, Figure 7: Please add a nomenclature. HKN, HKWF, HKAB…? Please explain two blocks. More discussion should be added to Section 3.3.1.
10. Figure 8: The same comments. Abbreviations are not clear to me. HKN, LCN…?
11. Add a summarizing sentence on what should be remembered after each section in results and discussion.
12. Conclusions: This section should be written by evaluating interesting results, and suggestions for future applications should be given. The format that concise text followed by bullets is recommended for the conclusions section.
Reviewer 2 Report
This paper investigated different fouling control strategies for high concentrated liquid desiccants concentrating of membrane distillation. It is a meaningful study and can be considered for publication after addressing the following concerns:
1. The background information is lengthy, while the advance in this research field is less introduced.
2. Line 101: The sentence is not expressed well and it reads strange. Besides, how is “high concentration” defined? Give the range.
3. What is the salt rejection rate of the commercial PVDF membrane?
4. What is the geometric parameters (eg, thickness, filament length, strand angle) of the used spacer?
5. Fig. 6: Why the initial flux of “membrane spacer” significantly higher than other groups? There is no fouling at the beginning. Besides, why the “air backwashing” group performed worse than “no cleaning”?
6. Fig. 7: the codes of the 8 groups should be explained in the figure caption.
7. Line 321: the sentence should be recast. UF is also a kind of pressure-driven membrane.
8. It is a pity that the authors did not conduct CFD simulation to analyze the mechanism of spacers on hydrodynamics and fouling control. The authors should have more in-depth discussion on this issue by either providing CFD simulation results, or citing some relevant references. The following references may not limit to MD process, but they focus on the effect of spacer geometry on hydrodynamics and fouling, which can contribute to the discussion.
https://doi.org/10.1016/j.watres.2020.116649
https://doi.org/10.1016/j.watres.2021.117146
https://doi.org/10.1016/j.desal.2021.114940
https://doi.org/10.1016/j.memsci.2022.120395
https://doi.org/10.1016/j.memsci.2022.121209
https://doi.org/10.1016/j.cej.2022.136563
9. Fig. 10: What does the green region mean? The green color is different from the mark of “F”? And why detect F element? There seems no F element in HCOOL and LiCl?
10. What is the value of the energy saved by the 3 methods respectively?
Round 2
Reviewer 1 Report
I have carefully reviewed the R1 version of the manuscript. The authors have responded to my comments by addressing my major concerns and have improved the manuscript accordingly. I have no comment at this stage.
Reviewer 2 Report
The concerns have been well addressed.